# A Comprehensive Transcriptomics Analysis Reveals Long Non-Coding RNA to Be Involved in the Key Metabolic Pathway in Response to Waterlogging Stress in Maize

**DOI:** 10.3390/genes11030267

**Published:** 2020-02-29

**Authors:** Feng Yu, Zengdong Tan, Tian Fang, Kaiyuan Tang, Kun Liang, Fazhan Qiu

**Affiliations:** 1State Key Laboratory of Biocatalysis and Enzyme Engineering, School of Life Sciences, Hubei University, Wuhan 430062, China; yufeng@hubu.edu.cn; 2National Key Laboratory of Crop Genetic Improvement, Huazhong Agricultural University, Wuhan 430070, China; zdtan1217@gmail.com (Z.T.); ft13026029102@163.com (T.F.); TKYcircle@163.com (K.T.); kl353739@126.com (K.L.)

**Keywords:** maize, waterlogging stress, mRNA, long non-coding RNA, co-expression network

## Abstract

Waterlogging stress (WS) in a dynamic environment seriously limits plant growth, development, and yield. The regulatory mechanism underlying WS conditions at an early stage in maize seedlings is largely unknown. In the present study, the primary root tips of B73 seedlings were sampled before (0 h) and after (2 h, 4 h, 6 h, 8 h, 10 h, and 12 h) WS and then subjected to transcriptome sequencing, resulting in the identification of differentially expressed protein-coding genes (DEpcGs) and long non-coding RNAs (DElncRs) in response to WS. These DEpcGs were classified into nine clusters, which were significantly enriched in several metabolic pathways, such as glycolysis and methionine metabolism. Several transcription factor families, including AP2-EREBP, bZIP, NAC, bHLH, and MYB, were also significantly enriched. In total, 6099 lncRNAs were identified, of which 3190 were DElncRs. A co-expression analysis revealed lncRNAs to be involved in 11 transcription modules, 10 of which were significantly associated with WS. The DEpcGs in the four modules were enriched in the hypoxia response pathways, including phenylpropanoid biosynthesis, MAPK signaling, and carotenoid biosynthesis, in which 137 DElncRs were also co-expressed. Most of the co-expressed DElncRs were co-localized with previously identified quantitative trait loci associated with waterlogging tolerance. A quantitative reverse transcription-polymerase chain reaction analysis of DEpcG and DElncR expression among the 32 maize genotypes after 4 h of WS verified significant expression correlations between them as well as significant correlation with the phenotype of waterlogging tolerance. Moreover, the high proportion of hypoxia response elements in the promoter region increased the reliability of the DElncRs identified in this study. These results provide a comprehensive transcriptome in response to WS at an early stage of maize seedlings and expand our understanding of the regulatory network involved in hypoxia in plants.

## 1. Introduction

Waterlogging is one of major abiotic stresses that limits the productivity of terrestrial plants and has increasingly and frequently occurred in recent years due to global climate change [1,2]. Oxygen deprivation is the key feature of waterlogging stress, as plants alter their metabolism, physiology, and development to acclimatize to the low levels of oxygen [3]. The cause of these changes is the dramatic reprogramming of gene expression and RNA molecules, including coding RNA (mRNA) and non-coding RNA (ncRNA) that has been quantified with the development of deep high-throughput sequencing [4,5]. Numerous investigations into *Arabidopsis* [6,7,8], rice [9], maize [10,11], poplar (*Populus* × *canescens*) [12], and cotton (*Gossypium hirsutum*) [13] have been conducted to elucidate changes in the gene transcription of mRNA in response to low oxygen or flooding stress; these studies have revealed the similarity in transcriptome responses among species [14,15]. However, knowledge of ncRNA involved in hypoxia responses is largely unknown.

Long non-coding RNA (lncRNA) is defined as a transcript that is longer than 200 nucleotides (nt) in length and lacks coding potential. These RNAs have also been reported to affect the expression of other genes [16]. Thousands of lncRNAs have been identified in many plant species, such as *Arabidopsis* [17,18], rice (*Oryza sativa*) [19,20], maize (*Zea mays* L.) [21], cotton (*Gossypium* spp.) [5], and castor bean (*Ricinus communis*) [22], with the rapid development of deep RNA-seq techniques. Recent studies have demonstrated that lncRNAs participate in diversified biological functions in different organisms [23,24]. Nevertheless, only a few lncRNAs have been well characterized in plants, including *COOLAIR* and *COLDAIR* in the regulation of flowering in *Arabidopsis* [25,26], *INDUCED BY PHOSPHATE STARVATION1*, which is involved in phosphate uptake [27,28]; *Enod40*, which is involved in nodulation [29,30]; *LDMAR*, which is required for pollen development in rice [31]; *LAIR*, which increases rice grain yield [32]; and *Ef-cd*, which shortens rice maturity duration without a yield penalty [33]. Remarkably, accumulated evidence has demonstrated the involvement of lncRNAs in biotic and abiotic stress responses in plants [34]. For example, the pathogen-responsive lncRNA *ALEX1*, which is highly induced by *Xanthomonas oryzae* pv. *Oryzae* infection in rice, has been shown to be involved in the jasmonic acid pathway and to offer resistance to bacterial blight [35]; a nucleus-localized lncRNA *DRIR* was found to be a positive regulator enhancing plant responses to drought and salt stress in *Arabidopsis* [36]; a total of 144 lncRNAs that respond to cadmium stress in rice at an early stage were identified to regulate the genes of cysteine-rice peptide metabolism in cis [37]; and nitrogen deficiency in *Populus* triggers 126 lncRNAs to change their expression and become involved in a complex regulatory network [38]. These findings collectively suggest the diversified functions of lncRNAs, as well as their importance in plant development and stress responses.

Hypoxia is a common phenomenon that occurs in plant tissues, especially in waterlogged organs. Studies on mammals have demonstrated that lncRNAs are key regulators involved in hypoxic stress, such as lncRNA *H19* protecting H9c2 cells against hypoxia-induced injury by targeting MicroRNA-139 [39] and contributing to hypoxia-induced CPC injury by suppressing Sirt1 [40], hypoxia-induced *HOTAIR* expression regulated by HIF1α (hypoxia-inducible factor) contributing to its roles in tumorigenesis [41], *TUG1* playing an important role in hypoxia-induced myocardial cell injury by regulating the miR-145-5p-Binp3 axis [42], and hypoxia-induced lncRNA *NUTF2P3-001* contributing to the tumorigenesis of pancreatic cancer by derepressing the miR-3923/KRAS pathway [43]. Little knowledge of lncRNA’s involvement in hypoxic stress in plant species has been acquired, with the exception of the observation that root-specific *AtR8* increases accumulation under hypoxic conditions [44]. Thus, the global identification and characterization of lncRNAs involved in hypoxia regulation in plants are needed.

Maize is an upland crop and sensitive to waterlogging stress (WS). To date, only three studies have focused on identifying the important regulators and processes involved in WS in maize through transcriptome sequencing (RNA-seq) analysis [10,11,45]. These studies focused on their central role in the adaptation process after long-term stress, but they did not provide a global view of the transcriptome at an early response stage. Remarkably, no experiment was performed to investigate the lncRNA response to WS in maize. In the present study, maize inbred line B73 was subjected to WS conditions at the second leaf stage, and the root tips of B73 were sampled before treatment (0 h, WS_0h) and after 2 h (WS_2h), 4 h (WS_4h), 6 h (WS_6h), 8 h (WS_8h), 10 h (WS_10h), and 12 h (WS_12h) to investigate the global expression of the protein coding gene (PCgenes) and lncRNAs at an early stage. Expression profiling and co-expression network analyses of PCgenes and lncRNAs were conducted to identify the key metabolic pathway and lncRNAs involved in WS responses. These results expand our understanding of the transcriptome response of PCgenes and the potential functions of lncRNAs under waterlogged root tips; these results also provide a valuable resource for elucidating the molecular regulatory network responding to WS conditions.

## 2. Materials and Methods

### 2.1. Plant Materials and Growth Conditions

Seeds of maize inbred line B73 were planted in a greenhouse with a controlled temperature (~28 °C/22 °C day/light cycle), a 14 h/10 h light/dark cycle (light time: 7:00 am–9:00 pm), and average humidity (60% average). The growth substrate and waterlogging treatments were conducted as previously described in [46]. Briefly, ten uniform seedlings were planted in a plastic pot, and treatments were applied by maintaining a 2–3 cm water layer above the substrate at the second leaf stage, and stress was treated at 8:00 am. To prepare the sample for sequencing, ten root tips (~2 cm) of the primary roots in each plot were collected before treatment (0 h, control) and after 2 h, 4 h, 6 h, 8 h, 10 h, and 12 h stress and mixed as one replication. Three replications were prepared at each time point. In total, 21 samples were snap frozen in liquid nitrogen and stored at −80 °C for total RNA isolation.

### 2.2. RNA Extraction, Library Construction, and Sequencing

Total RNA was extracted using TRIZOL reagent (Invitrogen, Gaithersburg, MD, USA) and purified with an RNeasy mini kit (QIAGEN, Germantown, MD, USA) according to the manufacturer’s instructions. RNA integrity was detected using gel electrophoresis. High-quality RNA was used for library construction using the Illumina TruSeq Stranded RNA Kit (Illumina, San Diego, CA, USA) following the manufacturer’s recommendations. Transcriptome sequencing of the prepared libraries was performed on an Illumina HisSeq 4000 system with paired-end 150-bp reads (Shanghai Personal Biotechnology, China).

### 2.3. Quantification and Standardization of the PCgenes and lncRNAs in the Transcriptome

A total of 21 RNA-Seq data sets were generated from this study. All raw reads that had been deposited into the SRA (accession number: SRP249592) and GEO (accession number: GSE146136) were assessed for quality using the program of FastQC (V0.11.3) and filtered using Trimmomatic (V0.38) [47] to obtain clean data. The clean reads of all samples were aligned to the maize B73 reference genome (Ref_Gen4) downloaded from MaizeGDB using Hisat2 (v2.0.5) [48] with the default parameters. The featureCount (V1.6.4) [49] in the Rsubread package was used for quantification and standardization to obtain the read count and TPM (transcripts per kilobase per million) value of each expressed gene, in which the TPMs represented the expression level of each transcript.

For identification of lncRNAs, the “bowtie2” (V2.3.4.1) [50] was applied to build index files. The assembly of each transcriptome datum was independently conducted using CUFFLINKS (V2.2.0) [51]; a merged.gtf file was obtained from merging these data together. “RSEM” (V1.3.0) [52] was utilized to normalize and calculate the TPM value. To identify high confidence lncRNAs, the Perl script LncRNA_Finder.pl was used to filter the lengths of the isoforms, the number of orfs, and Coding Potential Calculator 2 (CPC2) to eliminate potential protein-coding RNAs [21]. According to the annotation information of the merged.gtf file assembled by CUFFLINKS, the isoforms marked with u and x were filtered, and finally, the isoforms with an expression amount of 0 or a mean TPM < 0.5 under the same conditions were removed.

### 2.4. Differential Expression and Pathway Analysis

A differential expression analysis between pairs of samples was performed using the DESeq2 R package [53] to identify differentially expressed PCgenes (DEpcGs) and differentially expressed lncRNAs (DElncRs) for transcriptomes containing biological replicates, and the adjusted *p*-values were calculated using the Benjamini and Hochberg method [54] to control the false discovery rate. The standard for screening DEpcGs and DElncRs was set as 1) an adjusted *p* < 0.05 and 2) Log_2_ (a foldchange) > 1, or Log_2_ (a foldchange) < −1. The different expression profiles between the stress conditions (WS_2h, WS_4h, WS_6h, WS_8h, WS_10h, and WS_12h) and the normal condition (WS_0h) were analyzed. The mfuzz package [55] was used to divide the different time expression patterns into differential clusters. An enrichment analysis of DEpcGs was conducted using KEGG enrichment analysis. The online software AgriGO V2.O (http://systemsbiology.cau.edu.cn/agriGOv2/index.php) [56] was applied to analyze the gene ontology (GO). Heat maps and Venn diagrams were drawn using R packages.

### 2.5. Expression Network Construction

To investigate the co-expression profiling between PCgenes and lncRNA, all transcripts of highly-confident lncRNAs and DEpcGs were subjected to the weighted gene co-expression network analysis (WGCNA) method [57] to construct an expression matrix, and genes with similar expression patterns were clustered into the same module. The relationships between the transcripts in the module and the samples were investigated, and the important modules that were significantly associated with the sample traits (WS_0h, WS_2h, WS_4h, WS_6h, WS_8h, WS_10h, and WS_12h) were identified. The genes in the modules were then subjected to KEGG pathway enrichment. Finally, visualization of the co-expression network was performed using the Cytoscape (v3.5.0) [58].

### 2.6. Quantitative Real-Time (RT) PCR

The extracted total RNA of all samples subjected to transcriptome analysis was also used for quantitative real-time PCR. The total RNA was purified using RNase-free DNase (Invitrogen, Gaithersburg, MD, USA), and single-stranded cDNA was synthesized using recombinant M-MLV reverse transcriptase (Invitrogen) according to the manufacturer’s protocol. qRT-PCR was conducted using gene-specific primers (Appendix A) in a 25 μL reaction with a 2 × iTaq^TM^ Universal SYBR Green Supermix (BioRad, Hercules, CA, USA) under the following conditions: initial denaturation at 95 °C for 5 min, followed by 40 cycles at 95 °C for 15 s, 58 °C for 10 s, and 72 °C for 20 s. The internal reference *ZmActin1* was utilized to normalize the expression data. Relative expression levels were calculated according to the 2^−ΔΔCT^ (cycle threshold) method [59].

### 2.7. Verification of the Co-Expression Modules of DEpcGs and DElncRs in Different Inbred Lines

Thirty-two maize inbred lines in the association panel [48] were randomly selected and subjected to waterlogging stress during the second leaf stage, as mentioned above. The root samples of each line were collected before (0 h) and after (4 h) stress, and total RNA was extracted for further use. An expression analysis of DEpcGs and DElncRs was conducted using the qRT-PCR technique. The expression correlation between DEpcGs and DElncRs and significant *p*-values were calculated using the R software (R Development Core Team 2013; version 3.6.3; http://www.r-project.org/).

### 2.8. Conserved Motif Discovery

To discover the potential conserved motif in the promoter region of DElncRs, the DElncRs in each comparison were screened again based on stricter criteria: 1) log_2_ (a foldchange) > 1.5; 2) a baseMean of each comparison > 50; 3) *p* < 0.05. The 1500 bp upstream sequences in front of the transcript start sites of the screened DElncRs were downloaded from the B73 reference genome, and the resulting sequences were submitted to the PlantCARE database (accessed on: November 27, 2019) [60] to search for the conserved motif involved in stress and hormone response.

## 3. Results

### 3.1. The Time-Course Transcriptomic Profiles of Seedling Root Tips Exposed to WS Conditions

To investigate the gene expression dynamics of the root tips of the B73 seedling in response to WS, all the samples (WS_0h, WS_2h, WS_4h, WS_6h, WS_8h, WS_10h, and WS_12h) were subjected to transcriptome sequencing (RNA-seq). After high-depth RNA-seq, about one hundred million pair-end reads from each sample were generated, and the average rate of uniquely mapped reads aligned to maize reference genome [61] for all samples was 85.6% (Appendix A). Uniquely mapped reads were used to calculate the normalized transcription level as transcripts per kilobase per million (TPM), and the average TPM value > 1 of all the 21 samples of these genes were considered as an expression. In total, 23,175 expressed genes were detected in at least one sample (Appendix A).

For an investigation of the transcriptome differences of the PCgenes between normal (WS_0h) and stressed (WS_2h, WS_4h, WS_6h, WS_8h, WS_10h, and WS_12h) conditions, the differentially expressed PCgenes (DEpcGs) were analyzed (Appendix A). The up-regulated DEpcGs ranged from 1234 at WS-2h to 2732 at WS-6h and had similar gene numbers at WS-2h/WS-4h and WS-6h/WS-10h/WS-12h, respectively (Figure 1A). A total of 330 DEpcGs, including genes coding for enzymes involved in glycolysis and fermentation (*Zm00001d028759* encoded Pyruvate decarboxylase; *Zm00001d037689* encoded hexokinase7) and ethylene signaling related genes (*Zm00001d027622* encoded the C3HC4-type RING finger protein), were detected at all time points, and time point-specific differentially expressed genes (DEGs) numbered more than 70 in each sample. The down-regulated DEpcGs ranged from 555 at WS-2h to 2375 at WS-6h, which demonstrated the rapid growth of DEGs at WS-4h and a reduction at WS-8h (Figure 1B). These results suggest that the DEGs gradually accumulated in response to WS conditions in the early stage of stress (2 h, 1789 DEGs; 4 h, 3312 DEGs; 6 h, 5107 DEGs) and stably expressed during the late stage (10 h, 4654 DEGs; 12 h, 4506 DEGs).

To globally provide an expression profile after WS conditions, expression models based on a log_2_ (foldchange) of DEpcGs were created and were neatly divided into nine clusters (Cluster1–Cluster9), including 4261 genes ranging from 351 to 665 in each cluster (Figure 1C,D). The KEGG enrichment analysis demonstrated that three clusters, cluster1, cluster3, and cluster6, were significantly (FDR < 0.05) enriched in a specific pathway (Figure 1D). Four genes in Cluster1 (*Zm00001d048702* encoded indolin-2-one monooxygenase-like protein, *Zm00001d048703* encoded Cytochrome P450 71C1, *Zm00001d048705* encoded cytochrome P450 71C3, and *Zm00001d048710* encoded benzoxazinone synthesis 2) were enriched in benzoxazinoid biosynthesis, and 11 genes in Cluster2, including *Zm00001d052494* encoded pyruvate kinase and *Zm00001d034256* encoded phosphohexose isomerase 1, were enriched in glycolysis/gluconeogeneous (Appendix A). Cluster6 had the largest amount of DEpcGs and was enriched in four different pathways, including 24 genes (such as *Zm00001d022282*, *Zm00001d022283*, and *Zm00001d024752* encoded peoxidase) involved in phenylpropanoid biosynthesis, 9 genes (such as *Zm00001d017275*, *Zm00001d017276*, and *Zm00001d017279* encoded phenylalanine ammonia-lyase) involved in phenylalanine metabolism, 12 genes (such as *Zm00001d024850*, *Zm00001d024851*, and *Zm00001d024852* encoded 1-aminocyclopropane-1-carboxylate oxidase) involved in cysteine and methionine metabolism, and 10 genes involved in (α-) Linolenic acid metabolism (Appendix A). These significantly enriched metabolism pathways were tightly linked to the WS response. Moreover, a Gene Ontology (GO) analysis of Cluster1 to Cluster 9 also showed GO terms of the biological processes associated with WS conditions being significantly enriched in these clusters, except for Cluster9 (Appendix A). For example, the cell wall related processes in Cluster1 and Cluster2 (Appendix A), the energy metabolism-related process in Cluster3 (Appendix A), the regulation related process in Cluster4 (Appendix A), and the stress response-associated process in Cluster6 (Appendix A) were significantly enriched.

### 3.2. Differential Response of the Transcription Factor Families Involved in WS Conditions

Transcription factors play vital roles in regulating gene expression under normal and stress conditions. To investigate the characterization of the transcription factors (TF) involved in WS conditions, an enrichment analysis of all the TF families of maize downloaded from PlantTFDB was conducted, which revealed that 15 TF families were significantly enriched during at least one time point (Figure 2A). The enrichment of NAC, including four up-regulated genes, *Zm00001d000112*, *Zm00001d012527*, *Zm00001d013003*, and *Zm00001d019027*; MYB, including seven down-regulated genes such as *Zm00001d011669* and *Zm00001d035918*; and AP2-EREBP, such as *Zm00001d031728* and *Zm00001d040651*, were observed at six time points, and bHLH, bZIP, AUX-IAA, and SRS were enriched at at least three time points. A total of 45 NACs were differentially expressed after WS, about half of which were up-regulated (Figure 2B). Of the 48 bHLHs, 24 genes including five genes (*Zm00001d013130*, *Zm00001d027987*, *Zm00001d042263*, *Zm00001d043699*, and *Zm00001d04512*) at six time points were also up-regulated (Figure 2C). The up-regulated AP2-EREBP (Figure 2D) and bZIP including five up-regulated genes, *Zm00001d005962*, *Zm00001d0012296*, *Zm00001d017911*, *Zm00001d039065*, and *Zm00001d041920* (Figure 2F), were significantly enriched, whereas the down-regulated MYBs were also significantly enriched (Figure 2E). Investigations in plants have shown that group VII AP2-EREBP genes control flooding response and low-oxygen tolerance [62]. Of the 19 group VII AP2-EREBP genes in maize [46], eight genes were significantly up-regulated, implying that their potential functions regulate hypoxia response. These data demonstrated the discrepant performance of TFs under WS conditions.

### 3.3. Identification and Characterization of lncRNAs Responding to WS in the Root Tips of Maize Seedling

A total of 6098 transcripts (Appendix A) were finally identified as lncRNAs in B73 seeding root tips, including 5662 lincRNAs and 436 lncNATs (Appendix A). To clarify the characteristics of lncRNAs in maize root tips, the lengths, exon numbers, and expression levels of lincRNAs, lncNATs, and PCgenes were compared. The average lengths of the lincRNAs and lncNATs were 1044 nt and 1435 nt, respectively, which were shorter than PCgenes (3807 nt) (Figure 3A). Approximately 91% of the lincRNAs and 92% of lncNATs contained less than two exons, while only 42% of PCgenes had one or two exons (Figure 3B). Remarkably, about 29% of PCgenes had more than ten exons, whereas few lincRNAs and lncNATs were detected with this characteristic. Moreover, the normalized TPM demonstrated that the average TPM of PCgenes was higher than that of lincRNAs and lncRNAs (including lincRNAs and lncNATs) but lower than that of lncNATs (Figure 3C).

To identify the differentially expressed lncRNAs (DElncRs) that responded to WS conditions, the expression of lncRNA under stress was compared with expression under normal conditions (WS_0h). A total of 3190 DElncRs were identified in at least one stress condition (Appendix A), which included over half the proportion of identified lncRNAs, indicating that the expression level of lncRNAs was significantly induced by WS conditions. The up-regulated lncRNAs ranged from 373 at Ws_2h to 757 at WS_6h, and 788 lncRNAs were identified in more than two stress conditions (Figure 3D). The down-regulated lncRNAs had similar performance to up-regulated lncRNAs, and 811 lncRNAs were detected in more than two stress conditions (Figure 3E). These DElncRs were clearly classified into ten clusters based on a normalized log_2_(foldchange), in which the specific expression tendency of lncRNAs was presented. For example, Cluster2 was specifically up-regulated at WS_6h, and Cluster9 was specifically down-regulated at WS_4h (Figure 3F).

### 3.4. Association of the Expression Between lncRNAs and DEpcGs

To exploit the potential function of the lncRNAs involved in WS responses, a weighted gene co-expression network analysis (WGCNA) was performed. This analysis obtained 11 distinct modules shown in the dendrogram, in which major tree branches are labeled with different colors to highlight the different modules (Figure 4A). A total of 6035 lncRNAs were identified to be involved in 11 modules, ranging from one in the ‘purple’ module to 2677 in the ‘grey’ module (Appendix A), indicating that the lncRNAs participated in the DEpcG regulation network. The modules closely related to WS response were of particular interest in this present study. Thus, the correlations between the modules and distinct samples were calculated, and we found that all the modules except ‘pink’ were significantly (*p* < 0.05) associated with at least one of the WS samples (Figure 4B). An enrichment analysis of the DEpcGs in each WS-responded module was further conducted to identify the key metabolic pathways involving lncRNAs, and DEpcGs in five modules were significantly enriched, including ‘blue’ (EC2.7.11.1, MAPK signaling pathway), ‘brown’ (EC1.4.3.21, Phenylalanine metabolism; EC4.3.1.24, Phenylpropanoid biosynthesis), ‘red’ (EC2.5.1.99, Carotenoid biosynthesis), and ‘black’ (EC4.3.1.24, Phenylpropanoid biosynthesis), all of which play vital roles in the WS response. The DEpcGs and DElncRs involved in these pathways were screened for a further analysis according to their significant associations. For example, ‘blue’ was significantly related to WS_4h, and then the DElncRs at WS_4h were selected (Appendix A).

The ‘brown’ module was negatively related with WS_4h, and 22 DEpcGs and 35 DElncRs associated with the enriched pathway were down-regulated (Figure 4C). Most of these DEpcGs, including three genes (*Zm00001d003015*, *Zm00001d051166*, and *Zm00001d017276*) encoding phenylalanine ammonia-lyase and four genes (*Zm00001d024740*, *Zm00001d024753*, *Zm00001d007161*, and *Zm00001d008300*) encoding peroxidase. Over half of the DElncRs (such as *TCONS_00043110*, *TCONS_00077962*, *TCONS_00084669*, and *TCONS_00105920*) had high eigengene connectivity. The ‘blue’ group was positively associated with WS_4h; seven up-regulated DEpcGs were involved in the MAPK signaling pathway, and 64 DElncRs were up-regulated at WS_4h (Figure 4D). Of the seven DEpcGs, six genes (*Zm00001d012263* encoded the serine/threonine-protein kinase SAPK4, *Zm00001d016924* encoded the ETHYLENE INSENSITIVE 3-like 5 protein, *Zm00001d020939* encoded the EIL transcription factor, *Zm00001d037719* encoded the CAPIP1, *Zm00001d041920* encoded the transcription factor PosF21, and *Zm00001d047017* encoded putative HLH DNA-binding domain superfamily protein had higher connectivity to lncRNAs, whereas only a few DElncRs, such as *TCONS_00166326*, *TCONS_00060596*, and *TCONS_00149876*, had a higher connection. Only one DEpcG (*Zm00001d018819*, encoding nine-cis-epoxycarotenoid dioxygenase 5) was involved in carotenoid biosynthesis, and 28 down-regulated DElncRs at WS_4h were detected in the ‘red’ module (Figure 4E), which were connected with more than 15 DElncRs. The ‘black’ module was positively associated with WS_12h, and 12 up-regulated DEpcGs and 10 up-regulated DElncRs were identified (Figure 4F). All of these EDpcGs encoded peroxidase, and most of them had relative higher epigen gene connectivity than DElncRs, except *TCONS_00177501*. These results collectively demonstrated that the expression of lncRNAs was regulated in a similar way with PCgenes involved in these key pathways to respond to WS conditions.

### 3.5. The Expression of DElncRs Is Positively Associated with Waterlogging Tolerance

To verify the expression levels of DEpcGs and DElncRs in each module, 20 DEpcGs and 18 DElncRs were randomly selected to quantify their expression in B73 seedling root tips using qRT-PCR. The expression levels of DEpcGs and DElncRs were in agreement with the RNA-Seq data and the qRT-PCR experiment (Figure 5A, Appendix A), and significantly higher correlations between them were detected (Figure 5B,C). In order to analyze the co-expression profile of DEpcGs and DElncRs in the same module among 32 different genotypes, four pairs of DEpcGs and DElncRs (*Zm00001d029280* and *TCONS_00177501* in ‘black’; *Zm00001d012263* and *TCONS_00124833* in ‘blue’; *Zm00001d015618* and *TCONS_00105920* in ‘brown’; *Zm00001d018819* and *TCONS_00092298* in ‘red’) were chosen to analyze their expressions among the 32 maize inbred lines at WS_0h and WS_4h; the expression correlation among the phenotypes of survival rate (SR) after a long-term WS condition [48] for the DEpcGs and DElncRs in each pair was calculated. High correlations between DEpcGs and DElncRs, ranging from 0.58 to 0.72, were commonly displayed in the four modules, indicating that the co-expression of PCgenes-lncRNA occurred among different genotypes in response to WS conditions (Figure 5D, Appendix A). The expression of DEpcGs and DElncRs was also significantly correlated with the phenotype of SR, for which *Zm00001d029280* and *TCONS_00177501* in the “black” module were negatively correlated with SR, and the other three DEpcGs and three DElncRs in the other three modules were positively correlated with SR, suggesting their roles in waterlogging tolerance. These data further indicated that the expression of lncRNAs (such as *TCONS_00177501*) and PCgenes (such as *Zm00001d029280*) may coordinate the tolerance under waterlogging condition.

### 3.6. Most of the DElncRs Were Localized within the Previously Mapped Quantitative Trait Loci (QTL)

The DElncRs localized within known QTL regions could also be considered candidates to regulate waterlogging tolerance. We thus investigated the DElncRs involved in key WS-responding metabolic pathways in ‘blue’, ‘brown’, ‘red’, and ‘black’. A total of 137 DElncRs (Appendix A) were mapped to 10 chromosomes, and approximately 59% (81 of 137) of DElncRs were localized within the previously mapped QTL, of which only the DElncRs in chromosome 9 were not within any QTL regions (Figure 6). Moreover, about 47% (38 of 81) of co-localized DElncRs were mapped within more than two QTL regions. For example, the *TCONS_00092298* in chromosome 4 is located within three QTL [63,64,65] associated with six phenotype traits (PH, RA2nd, SDW, TDW, and RDW), *TCONS_00131183* in chromosome 5 is located within four QTL [29,63,66,67] controlling the phenotypes of PH and RA, and *TCONS_00089956* in chromosome 3 is located within three QTL [63,68] controlling the RA, GY, and RLod. These results demonstrate that DElncRs may also affect phenotype variations in response to WS.

### 3.7. Conserved Anoxic Motif in the DElncR Promoter

A total of 145 up-regulated DElncRs were ultimately identified to analyze their conserved motifs in the promoter region. The elements responding to three stresses were discovered; 88.3% (128 of 145), 43.4% (63 of 128), and 31.0% (45%) of these DElncRs had anoxic (anoxia response element, ARE; GC-motif), drought (MBS), and low-temperature (LTR) properties, respectively (Table 1). A few DElncRs had more than two anoxic elements; for example, *TCONS_00051666* had five ARE motifs, *TCONS_00034141* had four GC-motifs, and *TCONS_00043489* had one ARE and one GC-motif. Moreover, the responsiveness elements of abscisic acid, MeJA, auxin, gibberellin acid, and salicylic acid were also found in the DElncRs promoter, covering DElncRs ranging from 21.4% to 77.9%. These data suggest that the DElncRs were responding to stress and hormone stimuli, specifically under anoxic conditions.

## 4. Discussion

Investigating the mechanisms of gene regulation underlying WS conditions will contribute to a better understanding of the molecular basis for waterlogging tolerance and help maize adapt better to WS. Although the transcriptomic response involved in waterlogging conditions has been studied in previous works [10,11,45], there are few investigations focusing on early response, especially for the regulatory lncRNAs that participate in low-oxygen metabolism. The overall goal of the present study was to determine the transcriptional responses associated with the early stages of WS and characterize the role of the lncRNAs involved in key hypoxia-metabolic pathways via an RNA-Seq approach. Thus, the root samples of B73 seedlings subjected to WS for less than 12 h were collected, and the expressions of PCgenes and lncRNAs were cataloged, which revealed the important role of DElncRs in hypoxia-signaling in response to WS. The results of our comprehensive transcriptome analysis will help expand our knowledge on metabolic pathways during waterlogging in maize.

The major characteristic of waterlogged organs is their dramatically decreased concentration of oxygen, which leads to the alteration of the metabolic pathway from aerobic to anaerobic respiration [1,3]. The physiological responses, such as accumulated hydrogen peroxide will cause toxicity to plant cells, but the expressions of genes encoding antioxidant/metabolic enzymes are rapidly induced to maintain homeostasis. Signal transduction, protein degradation, ion transport, and carbon and amino acid metabolism played important roles during the late stage of WS (12–24 h) [11]. Furthermore, energy-production, programmed cell death, aerenchyma formation, and ethylene responsiveness were the main pathways adapted to long-term WS (5 days) [10]. The analysis of early responses to waterlogging in the present study also demonstrated that the expressions of a large number of genes changed, and DEpcGs were enriched in vital metabolic pathways (Figure 1, Appendix A). These DEpcGs were classified into nine clusters, among which three clusters (Clusters 1, 3, and 6) were significantly enriched in different KEGG pathways (Figure 1, Appendix A), and eight clusters were enriched in GO terms (Appendix A). The glycolysis/gluconeogenesis pathway, which produces energy and recycles carbon for other pathways to survive, was found to be up-regulated, indicating this pathway’s underlying central role in WS in early, late, and long-term responses. For example, six up-related genes, including *Zm00001d017121* encoded glyceraldehyde-3-phosphate dehydrogenase 4, *Zm00001d012103* encoded aldolase2, *Zm00001d034256* encoded phosphohexose isomerase1, *Zm00001d028759* encoded pyruvate decarboxylase isozyme 3, *Zm00001d037689* encoded hexokinase7, and *Zm00001d010588* encoded pyruvate decarboxylase1. The genes involved in glycolysis and fermentation in the present study were also up-regulated after five days WS in the tolerant line HKI1105 [10]. The phenylpropanoid biosynthesis pathway was significantly enriched, in which 22 genes were down expressed, including seven peroxidases (*Zm00001d007952*, *Zm00001d022282*, *Zm00001d022283*, *Zm00001d024752*, *Zm00001d037547*, *Zm00001d037550*, *Zm00001d046035*, and *Zm00001d050572*), which are enzyme scavengers for reactive oxygen species (ROS), demonstrating that ROS metabolism was also regulated during the early stage. Moreover, ethylene biosynthesis related genes, including six 1-aminocyclopropane-1-carboxylate (ACC) oxidase, such as *Zm00001d024843*, *Zm00001d024850*, and *Zm00001d024851*, which were up-regulated during the early stage, as well as one up-regulated ACC synthase (*Zm00001d026060*), were significantly enriched in the cysteine and methionine metabolism pathway. Ethylene signaling controls oxygen sensing under hypoxia conditions. A previous study showed that many genes were involved in the ethylene-response related pathway [10], in which nine genes, such as *Zm00001d027302* encoding the SKP1-like protein 21, *Zm00001d002896* encoding TUB-transcription factor 4, and *Zm00001d032849* encoding RING/U-box superfamily protein (Appendix A), adapted to long-term stress were up-regulated during the early response. These data collectively demonstrated that the hypoxia response pathways were activated during early stress conditions. Interestingly, four down-regulated genes involved in benzoxazinoid biosynthesis and 10 genes involved in the (α-) linolenic acid metabolism pathways were also significantly enriched, suggesting that they may play important roles in regulating early responses to WS.

Transcription factors, like DNA binding proteins, are involved in various growth, development, and stress response processes. Previous studies on *Arabidopsis* have reported that heat shock factors (HSFs), ethylene response-binding proteins, MADS-box proteins, the AP2 domain, leucine zipper, zinc finger, and WRKY factors increased in response to oxygen deprivation [6,7,60,61]. A comparative transcriptome analysis of tolerant and sensitive lines in maize and rice also demonstrated that ethylene response-binding proteins play a vital role in enhancing tolerance [9,45]. A total of 39 up-regulated AP2-EREBP genes, the major downstream components of ethylene signaling, were significantly enriched in the present study (Figure 2), among which eight genes belonged to group VII ERFs, implying the important roles of their responses to hypoxia in maize, as described in Arabidopsis and rice [62]. Moreover, a significant enrichment of the transcription factors of the bZIP, MYB, NAC, bHLH, AUX-IAA, LOB, HB, and GRAS genes was also detected (Figure 2), suggesting that multiple TF families are also directly involved in responses to low-oxygen in maize root tips.

Many studies have been conducted to investigate the transcriptomes of PCgenes in response to WS condition in plants such as *Arabidopsis*, rice, maize, cotton, and soybean [7,9,11,69,70]. However, there are few investigations centered on lncRNAs responding to WS. Indeed, lncRNAs are a diverse class of RNAs engaged in numerous biological processes [16]. With the development of high-throughput sequencing techniques, increasing numbers of functional lncRNAs have been identified, which, together with PCgenes, have been employed to reveal the high level of complexity of eukaryotic transcriptomes [71]. In this research, maize inbred line B73 was used to identify WS-related lncRNAs, and a total of 6099 high-confidence lncRNAs were identified in the roots, of which 5662 were lincRNAs and 436 were lncNATs; these quantities are far more than those reported by Li et al. [21], which may be due to the tissue specificity and updated reference genome of maize. Over half of these lncRNAs (3190) were differentially expressed under WS and the ten clear expression tendencies of DElncRs were clustered (Figure 3), indicating that lncRNAs with different catalogs directly responded to WS, which provides proof of lncRNA’s involvement in low-oxygen responses.

One of the main objectives of this study was to determine whether lncRNAs are involved in the regulation of key metabolic pathways in response to WS in root tips. A co-expression network analysis of DEpcGs and lncRNAs was conducted and showed that 10 of 11 modules were significantly associated with at least one WS trait (Figure 4), suggesting the potential association between PCgenes and lncRNAs in regulating hypoxia response. Remarkably, four modules (‘brown’, ‘blue’, ‘red’, and ‘black’) were significantly enriched in three metabolic pathways, including phenylpropanoid biosynthesis, MAPK signaling, and carotenoid biosynthesis, which are popular pathways that respond to hypoxia [1]; 137 DElncRs were involved in these pathways, indicating their vital roles in regulated hypoxia-related signaling. The association of DEpcGs and DElncRs was further verified among different maize genotypes (Figure 5), and the tight association between them clarified that DElncRs are co-expressed with DEpcGs in diverse lines. The expression of DEpcGs and DElncRs was also significantly associated with the waterlogging tolerant phenotype (Figure 5). Interestingly, most of these DElncRs were co-localized with previously identified QTL (Figure 6), suggesting that these DElncRs are the possible candidates to control the corresponding traits associated with waterlogging tolerance. Moreover, most (88.3%) of the 148 high-confident hypoxia response lncRNAs (highly induced by WS) had anoxic response elements, such as ARE and GT-box, in their promoters, providing the impression that hypoxia-induced lncRNAs have similar response methods to PCgenes, suggesting the possibility that they could manipulate DElncRs. These clues may imply that hypoxia-induced lncRNAs that were similar to the expression of PCgenes involved in hypoxia-related pathways also affect waterlogging tolerance in maize seedlings.

Plants have evolved the endogenous circadian clock to anticipate day and night changes, which has a profound influence on numerous biological processes such as gene expression, enzyme activity, metabolism, and stress response [72]. Accumulating evidence has demonstrated that core regulators of the circadian clock rhythm in *Arabidopsis*, *CIRCADIAN CLOCK ASSOCIATED 1 (CCA1) and LATE-ELONGATED HYPOCOTYL* (*LHY*) of morning-phase, are directly involved in the regulation of abiotic stresses such as cold response [73]. In our present study, *Zm00001d049543* encoding *CCA1* and *Zm00001d024546* encoding *LHY* had the highest expression at WS_0h and rhythmically down-regulated until WS_12h (Appendix A). These expression profiles were similar to *CCA1* in *Arabidopsis* under free-running condition during the daytime [74], implicating that WS imposed on B73 seedlings may not disrupt the rhythm of the circadian clock. Further investigations through a side-by-side time-point control would help us distinguish the expression difference caused by WS and the rhythm of the circadian clock. Nevertheless, our analysis presented here generated a relatively robust list of PCgenes and lncRNAs that respond to WS during the early stage in maize root tips, which will likely be useful for future functional genomics research and precipitate more comprehensive studies on gene regulation under hypoxia conditions. These findings contribute new knowledge to our understanding of the hypoxia regulatory network in plants.

## Figures and Tables

**Figure 1 genes-11-00267-f001:**
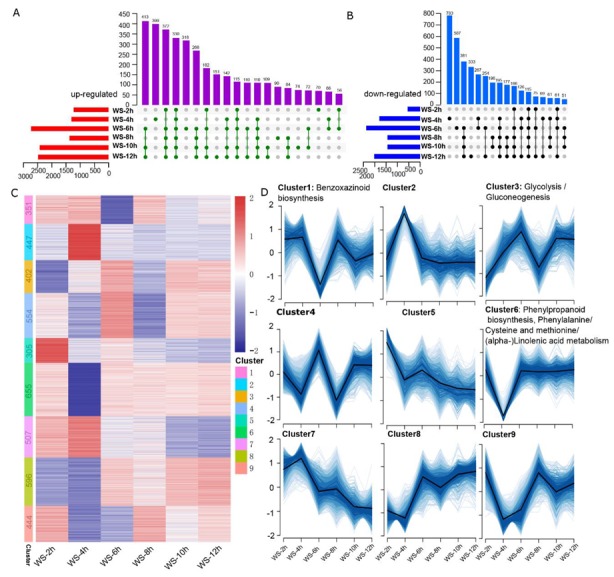
The characteristics of differentially expressed protein coding genes (DEpcGs) after waterlogging stress (WS). The Venn diagram shows the number of up-regulated DEpcGs (**A**) and down-regulated DEpcGs (**B**) after 2 h (WS_2h), 4 h (WS_4h), 6 h (WS_6h), 8 h (WS_8h), 10 h (WS_10h), and 12 h (WS_12h) of stress. These DEpcGs are divided into 9 clusters based on the ‘mfuzz’ package in the R software and were displayed using a heatmap (**C**) and their expression tendency (**D**). The numbers in the heatmap represent the DEpcG amounts in each cluster, and the significantly enriched KEGG pathways in Cluster1, Cluster3, and Cluster6 are shown.

**Figure 2 genes-11-00267-f002:**
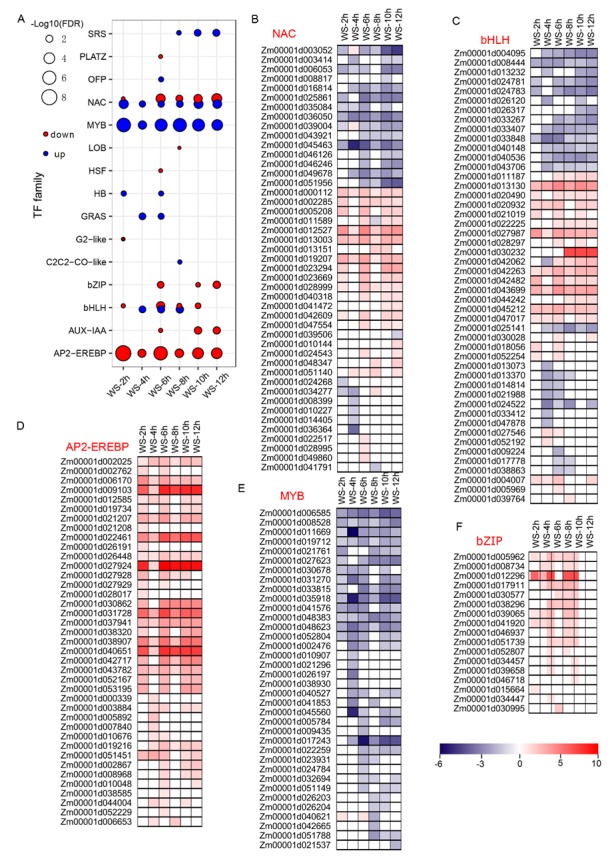
The enrichment analysis of transcription factors (TFs) after waterlogging stress (WS). The TF members of each family were downloaded from PlantTFDB (http://planttfdb.cbi.pku.edu.cn/). (**A**) The bubble chart shows the significantly enriched TF family in up- and down-regulated catalogs. (**B**) The gene list of NAC TFs enriched after WS treatment. (**C**) The gene list of bHLH TFs enriched after WS treatment. (**D**) The gene list of AP2-EREBP TFs enriched after WS treatment. (**E**) The gene list of MYB TFs enriched after WS treatment. (**F**) The gene list of bZIP TFs enriched after WS treatment.

**Figure 3 genes-11-00267-f003:**
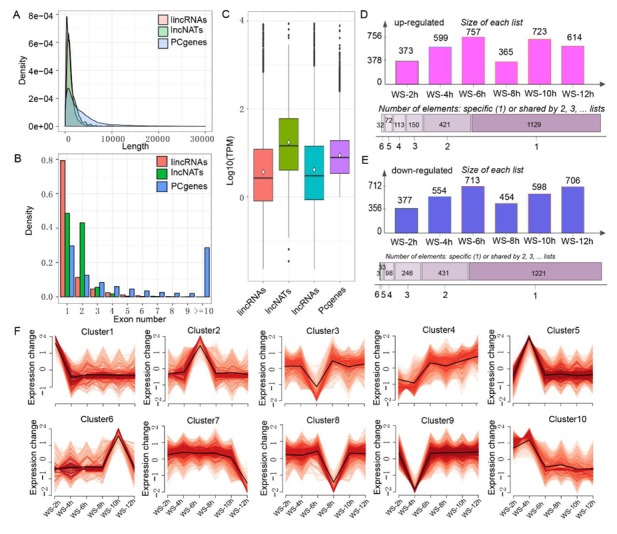
Characterizations of long non-coding RNA (lncRNAs) in maize root tips. (**A**) Length density distributions of long intergenic non-coding RNAs (lincRNAs), long non-coding natural antisense transcripts (lncNATs), and protein-coding genes (PCgenes). (**B**) Distributions of exon numbers in lincRNA, lncNATs and PCgenes. (**C**) The boxplot shows the expression level of lincRNAs, lncNATs, lncRNAs, and PCgenes. (**D**) The numbers of up-regulated lncRNAs after after 2 h (WS_2h), 4 h (WS_4h), 6 h (WS_6h), 8 h (WS_8h), 10 h (WS_10h), and 12 h (WS_12h) of stress. (**E**) The number of down-regulated lncRNAs in the samples of WS_2h, WS_4h, WS_6h, WS_8h, WS_10h, and WS_12h. (**F**) The expression clusters of differentially expressed lncRNAs based on the ‘mfuzz’ package in the R software.

**Figure 4 genes-11-00267-f004:**
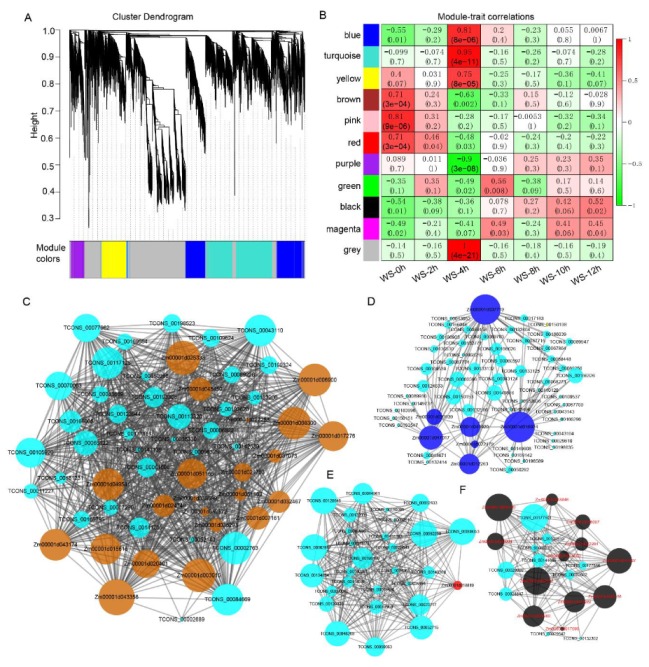
Co-expression network of the transcripts, including the protein-coding genes (PCgenes) and long non-coding RNA (lncRNAs) involved in waterlogging stress in maize root tips. (**A**) Hierarchical cluster tree and color bands indicating the 11 modules identified by the weighted gene co-expression network (WGCNA). (**B**) Analysis of the module–trait association. Each row represents a module, and each column represents a sample under waterlogging stress (WS) conditions. The numbers on the top and bottom of each cell represent the correlation and significant p-values, respectively. (**C**) The significantly enriched PCgenes (DEpcGs) involved in phenylpropanoid biosynthesis and the corresponding co-expressed differentially expressed lncRNAs (DElncRs) in the ‘brown’ module. (**D**) The significantly enriched DEpcGs involved in MAPK signaling and the corresponding co-expressed DElncRs in the ‘blue’ module. (**E**) The significantly enriched DEpcGs involved in carotenoid biosynthesis and the corresponding co-expressed DElncRs in the ‘red’ module. (**F**) The significantly enriched DEpcGs involved in phenylpropanoid biosynthesis and the corresponding co-expressed DElncRs in the ‘black’ module.

**Figure 5 genes-11-00267-f005:**
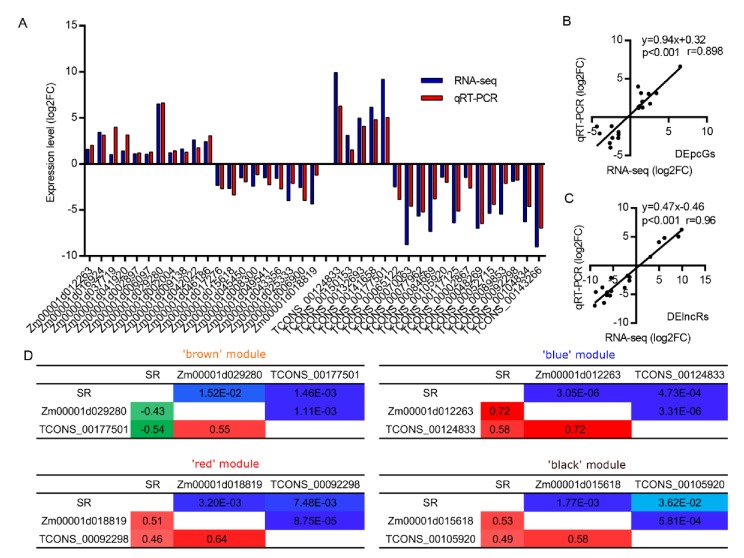
The expression levels of protein-coding genes (PCgenes) and long non-coding genes (lncRNAs) in maize seedlings after 4 h waterlogging stress (WS). (**A**) qRT-PCR verification of the expression of 20 differentially expressed PCgenes (DEpcGs) and 18 differentially expressed lncRNAs (DElncRs) in B73 seedlings after a 4 h WS condition. (**B**) The expression correlation between the qRT-PCR and RNA-seq of the 20 DEpcGs in B73 seedlings after a 4 h WS condition. (**C**) The expression correlation between qRT-PCR and RNA-seq of the 18 DElncRs in B73 seedlings after a 4 h WS condition. (**D**) The expression correlation of DEpcG and DElncR in the ‘brown’, ‘blue’, ‘red’, and ‘black’ modules among the 32 different maize genotypes after a 4 h WS condition are shown. The lower triangle indicates the correlation coefficient, and the upper triangle indicates the p-value of the correlation. The relative expressions of each DEpcG and DElncR in each genotype were calculated based on the control samples (not stress), which were further transformed through the function log_2_(). FC, foldchange; SR, survival rate of 32 maize genotypes, which were taken from Yu et al. (2019) [48].

**Figure 6 genes-11-00267-f006:**
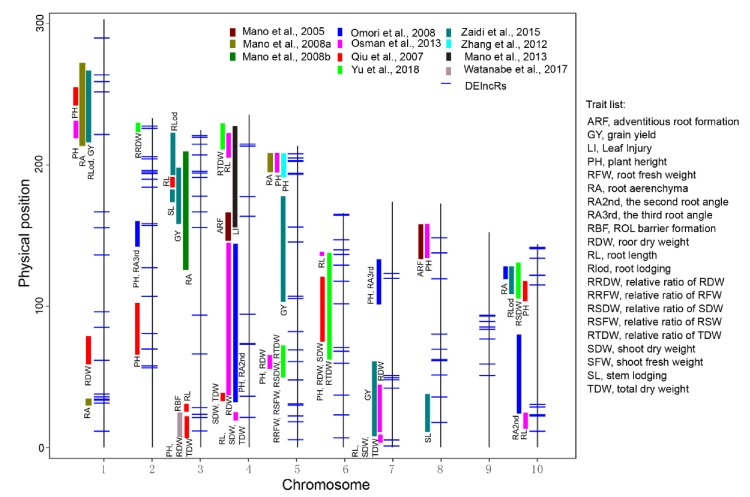
Co-localization of the differentially expressed long non-coding RNA (DElncRs) with previously identified quantitative trait loci (QTL) associated with waterlogging tolerance. The rectangular box represents the QTL interval, and the different colors indicate the different investigations.

**Table 1 genes-11-00267-t001:** The stress- and hormone-responsive elements in the 1.5 kb promoter region of the high-confidence hypoxia response of long non-coding RNA.

Responsiveness	Element	Number of DElncRs	Ratio (%)
Anaerobic	ARE, GC-motif	128	88.3
Drought	MBS	63	43.4
Low-temperature	LTR	45	31.0
Abscisic acid	ABRE	113	77.9
Auxin	AuxRR-core, TGA-element	63	43.4
Gibberellin acid	TATC-box, GARE-motif, P-box	75	51.7
MeJA	TGACG-motif	99	68.3
Salicylic acid	TCA-element	31	21.4

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
