# Peer review of "A Comprehensive Transcriptomics Analysis Reveals Long Non-Coding RNA to Be Involved in the Key Metabolic Pathway in Response to Waterlogging Stress in Maize"

_genes, 2020, doi:10.3390/genes11030267_

Round 1

Reviewer 1 Report

In their publication, Feng Yu and co-authors describe the transcriptional response of maize root tips to waterlogging for up to 12 h. The topic of that study is of interest for the flooding community. However, so far this manuscript only superficially describes expression differences of coding and non-coding RNAs, with no addition of physiological parameters. Below I have listed major problems in the current version of the manuscript.

One major problem is the lack of control time points. Plant metabolism and growth is strictly regulated by the circadian clock, and many diurnally regulated genes are known. One would at least expect a 2 h and a 12 h control time point. And if not for RNA seq, you should at least do those time points in qPCR with selected genes.

In line with this, please specify the day length in your experiments, the time of stress start relative to illumination start, and the illumination situation at the 12-h- time point.

Many references cited in introduction, especially regarding Arabidopsis transcriptomics, are very old, many newer ones exist.

Result section as well as discussion: you describe only gene families or functional categories in the text, and do not give any specific examples of differentially expressed genes. Also in the supplemental tables, the gene IDs need to be complemented with the annotation information. Right now, the tables are not helpful for any reader to find interesting genes.

Furthermore, all log2 ratios should be summarized in one data sheet, so one can compare expression directly over multiple time points. You could also compare your expression data with the published data from other time points in maize, and this information could also go into a supplemental table. You also should compare specific genes/ functions with expression data from other plant species under hypoxia/waterlogging, especially also in the discussion part.

RNAseq raw data need to be submitted to a public database like GEO or Array express. Please do so.

You mention groupVII ERFs (line 140), but not, why you specifically look at them. I know of their importance for hypoxia signaling, but this is not mentioned at all in your manuscript.

It is good that you also look into non-coding RNAs. However, you only do a correlation based on expression levels, which does not tell anything about their function. Can you be more precise or give examples of putative functions?

In line with this, you could by far not answer your major question that you mention in line 298. You only showed correlations, and not even correlations with tolerance (since you used 32 genotypes).

Line 336, can you mention the company or institute that did the sequencing?

Line 355, what input data did you use for DESeq2 analysis? You mentioned that you calculated TPM values, but the manual for the DESeq2 package explicitly mentions that one should not use normalized values, but raw reads/counts.

Line 112, a new paragraph would be helpful.

Line 264ff, you mention phenylpropanoid pathway to be differentially expressed, but not whether it is induced or reduced by the stress (I assume the second). This is only one example for a very superficial description of the data.

Figure 2, Figure S1, please use the same scale for all heatmaps.

Figure 5, I do not understand the values in Fig. 5A. Where are the PCR data for each of the 32 genotypes? I can only find one value per gene. Standard deviations and statistics are also missing for qPCR data.

Table 1, did you not find any HRPE motif in the dataset?

Overall, the text is difficult to read due to poor English style in many sections, especially in the introduction.

Author Response

1. One major problem is the lack of control time points. Plant metabolism and growth is strictly regulated by the circadian clock, and many diurnally regulated genes are known. One would at least expect a 2 h and a 12 h control time point. And if not for RNA seq, you should at least do those time points in qPCR with selected genes. R: Thank you for your comments. It was a constructive suggestion to consider the effect of circadian clock in our present study. As you said, we have not prepared the samples for RNA-seq under control condition at 2h to 12h. We are very pity that we could not completed this experiments at this time because the Coronavirus pneumonia outbreak in Wuhan and the laboratory was closed. In our previous study (Yu et al., 2019, Plant Biotechnology Journal), the qRT-PCR data of ZmEREB180 that plays important roles in hypoxia response has demonstrated that it was not affected by circadian clock. Thus, we speculated that hypoxia-responded genes may not seriously regulated by the circadian clock. We think this is a great topic to investigate the diurnally regulated genes under waterlogging condition in future study. 2. In line with this, please specify the day length in your experiments, the time of stress start relative to illumination start, and the illumination situation at the 12-h- time point. R: Thanks. In our experiment, the light/dark cycle was set as 14-h/10-h, and the light time was at 7:00 am to 9:00 pm. The treatment was conducted at 8:00 am, after 1 h illumination. Thus, all of the sampling process were operated under light condition. We have revised it in the manuscript accordingly. 3. Many references cited in introduction, especially regarding Arabidopsis transcriptomics, are very old, many newer ones exist. R: Thanks. We have renewed the two old references related to Arabidopsis transcriptomics as following: Van Veen et al., 2016, Plant Physiol; Vashisht et al., 2016, Plant Signal Behav. We have also renewed the Ref 16 (Ref 16: Quinn and Chang, 2016, Nat Rev Genet) and 19 (Ref19: Liu et al., 2019, BMC Genomics) and added a new reference in Ref47 (Gibbs et al., 2015, Plant Physiol). 4. Result section as well as discussion: you describe only gene families or functional categories in the text, and do not give any specific examples of differentially expressed genes. Also in the supplemental tables, the gene IDs need to be complemented with the annotation information. Right now, the tables are not helpful for any reader to find interesting genes. R: Thanks. We have added the specific examples of differentially expressed genes in Result and Discussion sections. And we have also complemented the annotation information in Table S2 and Supplementary Data Set S1. 5. Furthermore, all log2 ratios should be summarized in one data sheet, so one can compare expression directly over multiple time points. You could also compare your expression data with the published data from other time points in maize, and this information could also go into a supplemental table. You also should compare specific genes/ functions with expression data from other plant species under hypoxia/waterlogging, especially also in the discussion part. R: Thanks for your comments. All log2 ratios and pvalue of expressed genes were summarized in Table S2 along with the TPM value. In our study, we focus on the genes involved in short term response, and in Arora et al. (2017, Scientific Report), authors focused on the genes related to adaptive mechanisms. Thus, we can compare the expression data in these studies. However, we can not discover the list of all the expressed genes in the supplementary materials in Arora et al. (2017). We know we can download the raw data from NCBI to calculate the expression of these genes, we could not do this work at this time because of the Coronavirus pneumonia outbreak in Wuhan (Authors were separated in our hometown). On the other way, Arora et al. (2017) have listed the key differentially expressed genes involved in waterlogging stress, and we have added these information in Table S2. We also compared the specific genes/ functions with Arora et al. (2017). Please see the discussion section. 6. RNAseq raw data need to be submitted to a public database like GEO or Array express. Please do so. R: Thanks for your comment. We have summitted the RNAseq raw data to the SRA database with the accession number SRP249592, and the corresponding information has been added in the revised manuscript. 7. You mention groupVII ERFs (line 140), but not, why you specifically look at them. I know of their importance for hypoxia signaling, but this is not mentioned at all in your manuscript. R: Thanks. We have added the relative descriptions in the revised manuscript as following: “The investigations in plants have shown that group VII AP2-EREBP genes control the flooding response and low-oxygen tolerance [47]. Of the 19 group VII AP2-EREBP genes in maize [48], eight genes were significant up-regulated, implying their potential functions regulating hypoxia response.” 8. It is good that you also look into non-coding RNAs. However, you only do a correlation based on expression levels, which does not tell anything about their function. Can you be more precise or give examples of putative functions? R: Thank you for your comments. In our present study, we have calculated the association between PCgenes and lncRNAs and demonstrated that the co-expressions were appeared among them. We have analyzed the co-expression of DEpcGs and DElncRs from the same module in 32 different maize genotypes with the phenotype of survival rate (Yu et al., 2019, Plant Biotechnology Journal). The expression correlation between DEpcGs and DElncRs in each module was shown that they were significantly positively related, indicating that they co-expressed under waterlogging condition. In the revised manuscript, we have further calculated the correlation between the expression of DEpcGs and DElncRs and phenotypes of SR, and the results have demonstrated that these four DEpcGs and four DElncRs were significantly correlated with SR (Fig. 5D), indicating their roles in waterlogging tolerance. We have revised it in the result section, and we have also added the corresponding discussion to speculate their putative functions. 9. In line with this, you could by far not answer your major question that you mention in line 298. You only showed correlations, and not even correlations with tolerance (since you used 32 genotypes). R: Thanks. We have calculated the correlation between the expression of lncRNA and the phenotype of 32 lines under long-term waterlogging stress as description in the previous comment. And we have found that their expression were significantly associated with phenotype, indicating their roles in waterlogging tolerance. We have revised it in the manuscript correspondingly. 10. Line 336, can you mention the company or institute that did the sequencing? R: Thanks. We have added it in the revised manuscript. 11. Line 355, what input data did you use for DESeq2 analysis? You mentioned that you calculated TPM values, but the manual for the DESeq2 package explicitly mentions that one should not use normalized values, but raw reads/counts. R: Thanks. When the featureCount software were used, the reads count and TPM were calculated of each gene, in which the reads count was applied to quantify differentially expressed genes using DESeq2 R package. We have revised it in the manuscript correspondingly. 12. Line 112, a new paragraph would be helpful. R: Thanks. We have revised it based on your comment. 13. Line 264ff, you mention phenylpropanoid pathway to be differentially expressed, but not whether it is induced or reduced by the stress (I assume the second). This is only one example for a very superficial description of the data. R: Thank you for your comments. Most of the genes in phenylpropanoid pathway were reduced, and we have added more information in the result and discussion section in the revised manuscript. 14. Figure 2, Figure S1, please use the same scale for all heatmaps. R: Thanks. We have revised it. 15. Figure 5, I do not understand the values in Fig. 5A. Where are the PCR data for each of the 32 genotypes? I can only find one value per gene. Standard deviations and statistics are also missing for qPCR data. R: Thanks. We are sorry for our inexplicit descriptions in Fig. 5 legend. In fact, the data of qRT-PCR for 18 DEpcGs and 12 DElncRs were used to verify the accuracy of RNA-seq in B73 seedlings, and these results have been displayed in the result section “The expression of DElncRs positively associated with DEpcGs among different genotypes”. The qRT-PCR data for each of the 32 genotypes were used to analyze the co-expression module between DEpcGs and DElncRs. We have revised the legend of Fig. 5. Moreover, to compare the expression level and calculate the expression relationship between RNA-Seq and qRT-PCR data, the expression data of foldchange were further transformed through log2 function. Thus, Standard deviations and statistics were not provided. 16. Table 1, did you not find any HRPE motif in the dataset? R: Thanks. We have searched the HRPE motif in the promoters of 145 selected lncRNAs, and unfortunately HRPE motif was not discovered. 17. Overall, the text is difficult to read due to poor English style in many sections, especially in the introduction. R: Thanks. We have edited the language using MDPI English Editing Services.

Reviewer 2 Report

The authors have planned experiments to investigate the variation of trascription  in the early stage of maize seedling under waterlogging stress, using a trascriptomic approach.  The experiments are correctly performed and the "big data" obtained are correctly processed.The correlations observed are suggestive but not resolutive in the comprehension of the complex network regulating waterlogging stress. The authors can stress better this aspect. The experiments extend and complete previous research on late stages as cited by the authors and add the observations on lncRNAs. 

I think that a revision of english language by a mother-tongue may improve the quality of a manuscript. There are many typewriting errors scattered along the text. 

In the Supplemental Data set S1 the second column compared to WS0, after WS4 ,report wrong WS time treatment and must be corrected.

I have not found any comments on the 32 genotype used for the experiments of comparison between WS0 and WS4. 

Author Response

1. I think that a revision of english language by a mother-tongue may improve the quality of a manuscript. There are many typewriting errors scattered along the text. R: Thanks. We have edited the language using MDPI English Editing Services. 2. In the Supplemental Data set S1 the second column compared to WS0, after WS4 ,report wrong WS time treatment and must be corrected. R: Thanks for your comments. We have revised it. 3. I have not found any comments on the 32 genotype used for the experiments of comparison between WS0 and WS4. R: Thanks. These 32 genotypes used in the present study were from our previous study, which had the phenotype of waterlogging tolerance (Yu et al., 2019, Plant Biotechnology Journal). In the revised manuscript, we have added corresponding information, and further calculated the correlations between the expression of PCgenes/lncRNAs and phenotype.

Round 2

Reviewer 1 Report

The authors have provided a revision, which improved parts of the text and the supplemental material. However, the major concerns could not be addressed, and I still have a big problem with the lack of physiological data and subsequent experiments. Although I completely understand that the current situation in Wuhan is extremely difficult, this does not justify publishing premature manuscripts. At least the following things need to be addressed before publication:

If you cannot provide data on a 12-h-control time point, you need to add a comment on a possible clock-related gene expression difference in your dataset.

Although you submitted the raw files to the SRA archives, you should also do a submission of the mapped/calculated data to the GEO database.

Regarding non-coding RNAs: there is absolutely no evidence in your dataset that they are involved in regulation of gene expression. You only demonstrate a correlation of their expression, which indicates that their expression is regulated in a similar way, but by no means that they influence other genes. This statement would require genetic works (i.e., manipulation of lncR levels and study of its effects).  Please modify the statements in lines 222, 241, 255, 360.

Figure 5, you need to give more data here. Please provide two more supplemental tables, one should show qPCR data of B73 with statistics (FC and SD), and one should show qPCR data (FC and SD + statistics) for the 32 maize lines. And since the cited maize collection [48] used more than 32 lines, you need to add line IDs, origin and SR index to this table. From that cited publication, there are no data available about the line identities or a list with SR for each line. In line with this, line 439 does not refer to the correct paper.

Line 110f (and 297f), pyruvate decarboxylase and hexokinase are not energy-producing. Please rather use the phrase "genes coding for enzymes involved in glycolysis and fermentation".

Line 142, you analyzed not only one TF family, but families.

The text editing service did a good job, but the newly written parts still contain problematic phrases, for example lines 300, 375.

Author Response

  1. If you cannot provide data on a 12-h-control time point, you need to add a comment on a possible clock-related gene expression difference in your dataset.

R: Thank you for your comment. We have added the comment on the expression of core morning-phase genes (Zm00001d049543 encoding CCA1 and Zm00001d024546 encoding LHY). Please see the last paragraph of discussion.

  1. Although you submitted the raw files to the SRA archives, you should also do a submission of the mapped/calculated data to the GEO database.

R: Thanks. We have been uploading the gene expression matrix file to the GEO database based on your comment. However, transmission of rawdata has not been completed due to the problem of network speed, and about 50% rawdata have not been transmitted so far although over 100 hours were used. We have also ask for GEO service whether the raw data was shared with SRA database, unformately we didn't get an email reply from GEO. Thus, within five days of revised time, we could not completed the data submission. On the other hand, we have uploaded the raw data into SRA database with detailed description and the expression matrixs of PCgenes and lncRNA have been listed in supplementary materials (Table S2 and Table Table S3, TPM values). We think that these data have contain the information in GEO database. Thank you again for your rigorous attitude. However, we did not disrupt the submission process, and we will provide the GEO accession number if it has finished.

  1. Regarding non-coding RNAs: there is absolutely no evidence in your dataset that they are involved in regulation of gene expression. You only demonstrate a correlation of their expression, which indicates that their expression is regulated in a similar way, but by no means that they influence other genes. This statement would require genetic works (i.e., manipulation of lncR levels and study of its effects).  Please modify the statements in lines 222, 241, 255, 360.

R: Thank you for your comment. We have revised it.

  1. Figure 5, you need to give more data here. Please provide two more supplemental tables, one should show qPCR data of B73 with statistics (FC and SD), and one should show qPCR data (FC and SD + statistics) for the 32 maize lines. And since the cited maize collection [48] used more than 32 lines, you need to add line IDs, origin and SR index to this table. From that cited publication, there are no data available about the line identities or a list with SR for each line. In line with this, line 439 does not refer to the correct paper.

R: Thanks. We have added two supplemental tables (Table S8 and Table S9) based on your comments. We have also revised the cited reference for the SR phenotype.

  1. Line 110f (and 297f), pyruvate decarboxylase and hexokinase are not energy-producing. Please rather use the phrase "genes coding for enzymes involved in glycolysis and fermentation".

R: Thanks. We have revised it.

  1. Line 142, you analyzed not only one TF family, but families.

R: Thanks. We have revised it.

  1. The text editing service did a good job, but the newly written parts still contain problematic phrases, for example lines 300, 375.

R: Thanks. The text editing service have edited all the content including our newly written parts. We have also carefully checked the manuscript to avoid the problematic phrases.

Round 3

Reviewer 1 Report

Revised changes are fine

upon successful submission, add GEO ID